# Pain and Opioid-Induced Gut Microbial Dysbiosis

**DOI:** 10.3390/biomedicines10081815

**Published:** 2022-07-28

**Authors:** Karen R. Thomas, Jacob Watt, Chuen Mong J. Wu, Adejoke Akinrinoye, Sairah Amjad, Lucy Colvin, Rachel Cowe, Sylvia H. Duncan, Wendy R. Russell, Patrice Forget

**Affiliations:** 1School of Medicine, Medical Sciences and Nutrition, University of Aberdeen, Foresterhill, Aberdeen AB25 2ZD, UK; u18jw20@abdn.ac.uk (J.W.); u05cw20@abdn.ac.uk (C.M.J.W.); u17aa20@abdn.ac.uk (A.A.); u12sa20@abdn.ac.uk (S.A.); l.colvin.20@abdn.ac.uk (L.C.); u25rc20@abdn.ac.uk (R.C.); sylvia.duncan@abdn.ac.uk (S.H.D.); w.russell@abdn.ac.uk (W.R.R.); patrice.forget@abdn.ac.uk (P.F.); 2Rowett Institute, University of Aberdeen, Foresterhill, Aberdeen AB25 2ZD, UK; 3Epidemiology Group, Institute of Applied Health Sciences, University of Aberdeen, Foresterhill, Aberdeen AB25 2ZD, UK; 4Department of Anaesthesia, NHS Grampian, Foresterhill, Aberdeen AB25 2ZD, UK

**Keywords:** opioids, opioid-induced dysbiosis, OID, dysbiosis, opioid induced hyperalgesia, gut homeostasis, faecal microbiota transplantation, gut-brain axis

## Abstract

Opioid-induced dysbiosis (OID) is a specific condition describing the consequences of opioid use on the bacterial composition of the gut. Opioids have been shown to affect the epithelial barrier in the gut and modulate inflammatory pathways, possibly mediating opioid tolerance or opioid-induced hyperalgesia; in combination, these allow the invasion and proliferation of non-native bacterial colonies. There is also evidence that the gut-brain axis is linked to the emotional and cognitive aspects of the brain with intestinal function, which can be a factor that affects mental health. For example, *Mycobacterium*, *Escherichia coli* and *Clostridium difficile* are linked to Irritable Bowel Disease; *Lactobacillaceae* and *Enterococcacae* have associations with Parkinson’s disease, and *Alistipes* has increased prevalence in depression. However, changes to the gut microbiome can be therapeutically influenced with treatments such as faecal microbiota transplantation, targeted antibiotic therapy and probiotics. There is also evidence of emerging therapies to combat OID. This review has collated evidence that shows that there are correlations between OID and depression, Parkinson’s Disease, infection, and more. Specifically, in pain management, targeting OID deserves specific investigations.

## 1. Introduction

Acute (and thus persistent) pain typically begins with nociceptors; the terminal ends of sensory neurons which are found within the peripheral nervous system (PNS) and are often managed with opioids [1,2]. High-threshold primary sensory neurons respond to potential damage to the body by transmitting the painful stimuli to the second-order neurons in the dorsal horn of the spinal cord. The signal is then carried up the spinothalamic tract to the thalamus and somatosensory cortex [2]. Opioids induce their effects in the body by binding to G protein-coupled receptors, of which there are four types (mu, delta, kappa, and nociception) [2]. Endogenous opioid peptides are classified into four groups: β-endorphins, enkephalins, dynorphins and nociception/orphanin FQ [3]. Opioid agonists can activate both central and peripheral opioid receptors, with morphine being the most commonly used exogenous opioid analgesic [4].

Opioids are potent analgesics that are inexpensive and commonly prescribed to manage pain around the world [5]. Opioid receptors are expressed in the PNS, the GI tract and the enteric nervous system’s (ENS) myenteric plexus, even though the direct pharmacological effects of opioids are dependent on opioid receptor distribution in the central nervous system (CNS) [5]. The gastrointestinal (GI) system consists of the organs that form the gastrointestinal tract from the mouth to the anus. The central nervous system (CNS) consists of the brain (encephalon) and the spinal cord (medulla spinalis) and is the body’s processing centre.

Patients may have not just constipation but also hyperalgesia and immunosuppression, which may be clinically significant in some opioid-treated patients. Intriguingly, modifications in the gut microbiota have been reported in individuals who have experienced similar issues due to opioid therapy.

Gut microbial dysbiosis is a change in the gut microbiota’s functional or structural configuration that disrupts gut homeostasis and is linked to several diseases [5]. The changes in the balance and composition of gut microbiota are referred to as opioid-induced dysbiosis (OID), and they are linked to a variety of disease states and the development of antinociceptive tolerance.

A significant amount of new research suggests that gut microbiota has a significant impact on the health of individuals. Gut microbiota changes clearly impact intestinal homeostasis, the gut microbiome, the immune system, physiology, and host metabolic pathways [6].

Dysbiosis reduces microbial diversity and, therefore, alters the expression and transmission of neurotransmitters, affecting both the CNS and gut functions [7]. However, whether gut microbiota modifications may mediate unwanted effects related to opioids has not been extensively explored.

The aim of this review is to highlight any possible unidentified or potential links between opioids and gut microbial dysbiosis, the implications these may have, and, finally, to consider some promising research avenues.

## 2. The Pain Pathway and Hyperalgesia

The classic three-neuron chain nociceptive pathway is now understood to be a dual system at each level. The sensation of pain is thought to arrive in the central nervous system (CNS), with the discriminative component of pain (“first pain”) carried separately from the affective-motivational component of pain (“second pain”) [8]. In addition to spinal control mechanisms for nociceptive transmission, descending pathways from significant areas of the brain—the cortex, thalamus, and brainstem—can modify spinal functions. The CNS can respond to painful stimuli at multiple levels involved in pain transmission, modulation, and sensation [9].

Local circuits in the dorsal horn of the spinal cord play an essential role in processing nociceptive afferent information and influencing the actions of descending pain modulation systems. Opioids have inhibitory effects on target neurons in the short term, while stimulating effects have also been described. Noxious stimulation enhances the neuronal activity and alters gene expression, including immediate-early genes and neuropeptide (i.e., opioid) genes at the spinal and supraspinal levels of the somatosensory system.

### 2.1. Opioid-Induced Hyperalgesia

Opioid-induced hyperalgesia (OIH) is a state of nociceptive sensitization caused by the exposure to opioids. It is characterised by a paradoxical response whereby a patient receiving opioids to treat pain could become more sensitive to certain painful stimuli. Due to central sensitization long-term opioid treatment has the same neuroinflammatory potential responsible for pain chronicity, resulting in paradoxical worsening rather than pain relief [10]. 

Neuroinflammation is a unique type of inflammation occurring in response to noxious stimuli in the peripheral and central nervous systems [10]. It begins with altered vascular permeability, followed by leukocyte recruitment and the activation of microglia and astrocytes in the spinal cord and CNS. In many painful states, this mediates and may worsen inflammatory pain. The type of pain experienced might be the same or different from the underlying pain.

According to recent research, cholecystokinin is upregulated in the rostral ventromedial medulla (RVM) during chronic opioid exposure. Cholecystokinin has anti-opioid and pronociceptive properties, activating descending pain facilitation mechanisms from the RVM, enhancing nociceptive transmission at the spinal cord and promoting hyperalgesia. The neuroplastic adaptive changes induced by opioid exposure promote increased pain transmission, resulting in decreased antinociception (i.e., tolerance) [11,12,13]. 

### 2.2. Experimental Evidence

To explore OIH in human volunteers, several experiments have been carried out [14,15,16] (such experiments looked at the effect of short-term opioid infusion on an experimental stimulus that had been rendered hyperalgesic prior to the start of drug infusion [14]. The second set of studies looked at the effects of opioid antagonist-induced withdrawal on cold pressor pain in volunteers who had become acutely exposed to opioids. Several studies found that a 30 to 90-min infusion of the ultra-short-acting opioid agonist, remifentanil, aggravated pre-existing mechanical hyperalgesia. Aggravation was reflected by a 1.4 to 2.2-fold increase in the hyperalgesic skin area compared to preinfusion measurements. The magnitude of this effect was directly related to the duration of the infusion and the opioid dose. Aggravation of pre-existing hyperalgesia was observed up to four hours after stopping the remifentanil infusion but was no longer present the next day. These changes are consistent with an expanded area of secondary hyperalgesia and the result of enhanced nociceptive signal processing at the spinal cord level [14]. 

The central glutaminergic system has been proposed as a common mechanism for OIH development [17]. The excitatory neurotransmitter, N-methyl-D-aspartate (NMDA), may play a role in developing OIH in this system. Silverman^14^ outlined the role of NMDA in the development of OIH in a review.

The review highlights how NMDA receptors become activated, and inhibition prevents the development of tolerance and OIH. When the glutamate transporter system is inhibited, there is an increase in glutamate available to NMDA receptors. As a result, there may be a cross-talk of neural mechanisms of pain and tolerance. In the dorsal horn, prolonged morphine administration induces neurotoxicity via NMDA receptor-mediated apoptotic cell death.

These findings point to a mechanism in which inhibiting the NMDA receptor prevents OIH. This NMDA-mediated mechanism sensitizes neurons via the central glutaminergic system and may explain how OIH develops. Spinal dynorphins may also contribute to OIH by increasing the presence of excitatory neuropeptides, which can improve nociceptive input [17]. 

### 2.3. Antinociceptive Tolerance to Opioids

Given the clinical and social implications of physical dependence and addiction, the mechanisms causing opioid tolerance have been studied for many decades. Despite multiple studies at various levels in different tissues, no single regulatory mechanism can account for all of the variations observed in tolerance development [18]. 

Antinociceptive tolerance is clinically manifested as decreased or lost pain relief from a given opioid dose administered repeatedly or with continuous administration of an opioid over a period of time. The need for increasing opioid doses in chronic pain cases is well documented and is typically presented as a major barrier to providing adequate pain relief over a long period of time [11]. 

The chronic administration of morphine has recently been shown to alter the gastrointestinal tract’s resident microbial composition and induce bacterial translocation across the gut epithelial barrier via mechanisms involving toll-like receptors (TLR) 2 and 4 [18].

The primary mechanism for tolerance to the antinociceptive effects of chronic morphine appears to be dysbiosis of the gut microbiome. The precise mechanism by which the gut microbiome influences tolerance development is unknown; however, it is possible that the breakdown of the epithelial barrier, microbial translocation, and inflammation within the colonic wall may alter extrinsic sensory neurons, resulting in tolerance development [18]. 

## 3. The Gut-Brain Axis, Withdrawal and Diseases

The gut-brain axis is a bidirectional communication pathway between the intestines and the gut, linking emotional and cognitive aspects of the brain with intestinal function [19]. It is primarily controlled by intestinal microbiota and opioid receptors. This plays a vital role in disease progression when affected by dysbiosis. In neurodegenerative disorders such as Parkinson’s and depression or functional gastrointestinal disorders, evidence shows that enteric microbiota interacts locally with the intestines and directly with the CNS.

For example, the secretion of 5-hydroxytryptamine (5-HT) by enterochromaffin cells can activate afferent nerve endings to communicate to the CNS, even though 5-HT has intrinsic roles in metabolic management within the gut and peripherally. Short-chain fatty acids have recently been shown to regulate enterochromaffin cell production of gut-derived 5-HT [20]. Furthermore, 5-HT has significant links to psychiatric disorders, with the most commonly used antidepressant, fluoxetine, blocking the transport of gut 5-HT into plasma. Young male adults with autism spectrum disorder have shown an inverse relationship between high plasma serotonin and low serotonergic neurotransmission [20]. 

### 3.1. Addiction

Addiction is a severe form of a substance use disorder; patients show dependence and inability to cease use, no matter how badly they wish to. Addiction results in increased dopamine levels and abnormal signal imbalance in dopamine receptors to reinforce the ‘reward’ system created—a pleasurable response to stimuli to hook the subject further [21]. Opiate usage leads to a pro-inflammatory response in the gut, with a marked increase in Proteobacteria, contributing to addiction, as shown through the modulatory and reversible effect afforded by anti-inflammatory cytokines [22]. It is essential to consider that not all opiate users are affected by addiction, and the ones that are show higher sensitivity to drug effects due to genetic, molecular and behavioural factors, however opioid use in itself is a factor that increases sensitivity to addiction [23]. 

### 3.2. Opioid Withdrawal

The amygdala is the primary centre of the brain implicated in both opioid dependence and withdrawal effects; it receives information from the vagus nerve, which is affected in gut dysbiosis [22]. Opioid withdrawal may be due to the sudden cessation of opioids or if a patient is given a partial antagonist. Morphine withdrawal can also cause spontaneous sepsis and immunosuppression, leaving patients at risk of various infections [24]. A study taken on various cells in the CNS proved withdrawal to be an effect of dysbiosis, as a lower ratio of *Firmicutes* to *Bacteroides* was observed [22]. Opioid withdrawal severity can be measured using the Clinical Opioid Withdrawal Scale, which marks against signs and symptoms of opioid withdrawal. When many indicators are present, pharmacological management must be implicated, usually with methadone, which may be given as an outpatient or inpatient prescription [25]. 

### 3.3. Inflammatory Bowel Disease

Dysbiosis in intestinal microbiota is a critical factor in the pathogenesis of various diseases, one of the most prominent being inflammatory bowel disease (IBD). While IBD is a polyfactorial disease, microbial imbalance results in the increased prevalence of *Mycobacterium*, *Escherichia coli* and *Clostridium difficile*, all considered major pathogens of IBD [26]. However, further research must be undertaken, as it is not clear whether the dysbiosis viewed is a risk factor or an effect of IBD [1]. It is likely that microbial dysbiosis is the precursor to IBD. This is because the bacterial species, which are reduced in abundance, primarily serve luminal defence functions; including the butyrate-producing *Firmicutes*. Therefore, the gut becomes more sensitive and is likely to become inflamed with a lower quantity of these bacteria [26]. 

### 3.4. Parkinson’s Disease

Parkinson’s disease is caused by the degradation of nerve cells in the substantia nigra in the brain, resulting in dopamine underproduction. The link between Parkinson’s and the gut-brain axis has been studied extensively, showing that patients have a higher abundance of *Lactobacillaceae* and *Enterococcacae* (Table 1) [27]. Levodopa is the treatment of choice for Parkinson’s and increases opioid transmission, which causes patients to have dyskinesia, which is an impairment of voluntary movement. It has not been proved definitively that gut dysbiosis is a precursor to Parkinson’s. However, changes in the microbiota lead to neuroinflammation, which may affect the course of the disease and the response to medication [28].

The possibility that Parkinson’s may arise from the gut is supported by the fact that patients have a specific faecal microbiota makeup, even when on the same diet as other, healthy members of their household. A study on mice showed that dysbiosis caused the intestinal barrier to become hyperpermeable and levels of anti-inflammatory markers decreased changes that are mimicked in the brains of patients with progressive Parkinson’s disease [29].

### 3.5. Gut Function and Mental Health

The gut and brain act in a symbiotic relationship—as much as the gut can influence the brain, the brain can also influence the gut [30,31]. Much of the research focuses on how changes to the gut may influence mental health and vice versa. Furthermore, how the gut’s microbiome specifically improves or worsens depression has been revealed.

There is some evidence showing how the activation of MORs and DORs can influence mood and mental health by influencing the expression of neurotransmitters in the brain [32]. This shows that opioid medications can cause or exacerbate depression in such patients. There is no clear research showing a correlation between opioid abuse and depression. There is evidence showing that opioids alter the microbiome and that the microbiome affects mental health symptoms, but evidence has not extended to draw conclusions about the direct result of OID on depression. It may be that the mechanism of the opioids changing the bacterial composition of the gut acts as a critical pathway in mental health symptoms associated with these drugs.

Bacteria are crucial metabolizers in our gut. Some of their products are neurotransmitters, including serotonin [33], which is central to the proposed mechanism of depression [34]. The abundance of certain bacteria seen in microbial dysbiosis, potentially due to opioid use, depletes serotonin levels. One of the classes of bacteria prevalent with OID is *Alistipes*, a type of indole positive bacteria [34] which metabolizes tryptophan by hydrolyzing tryptophan to an indole molecule. Tryptophan is hydrolysed to 5-hydroxytryptamine, which is further decarboxylated to serotonin [35]. Therefore, the presence of *Alistipes* results in the reduction of serotonin levels and other tryptophan-derived molecules by reducing available tryptophan, and consequently increases rates of depression [34].

However, it is not as simple as suggesting that changes to the microbiome cause depression. Other bacteria show increased prevalence in the dysbiotic gut, such as the *Bifidobacterium* genus [22], which reduces depression [36]. *Bifidobacterium* increases brain-derived neurotrophic factor (BDNF), reduces monoamine oxidases and reduces oxidative stress [20]. BDNF is another neurotransmitter that is reduced in depression. Its use in probiotic yogurt resulted in clinically reduced levels of depression [37]. While the exact mechanism is not clear, it can be assumed that this genus would reduce depression by increasing the availability of serotonin and other neurotransmitters. Therefore, more consideration needs to be given to how opioids influence depression, as this evidence suggests there are beneficial and negative consequences in relation to mood.

Another pathway that has been suggested considers the effect of pro-inflammatory substances released by bacteria or by the presence of certain bacteria, which can then spread and lead to mental health disorders. The proliferation of Proteobacteria in the gut can enhance the dysbiotic state [38], and promotes inflammation via lipopolysaccharide production, which passes into the brain, causing increased levels of Tumour Necrosis Factor-α and reduced BDNF [37]. Both of these contribute to depression symptoms [39]. This can be further exacerbated as inflammation causes increased permeability in the gut blood barrier via Interleukin-17A and Interferon-γ [40] as well as the blood-brain barrier [41]—and therefore the increased movement of inflammatory substances around the body and into the brain. This inflammation seen through the activation of cytokines, interleukins and other substances affects several areas of the CNS [40].

An additional area that could be explored is the effect of opioid dysbiosis, specifically on the hypothalamic-pituitary axis (HPA) [42]. Dysbiosis can be caused by disruption to the HPA [31], and as this is known to be a two-way communication [43], it would be possible that dysbiosis itself could cause impairment to the HPA. Disruption to the HPA is frequently reported in patients suffering from depression [44,45] and this, combined with neuroinflammation, is highly indicative of depression. The mechanism of the HPA and depression utilises cortisol, a stress hormone. If changes to cortisol levels were noted with opioid dysbiosis, this could be an area of further study.

## 4. Gut Homeostasis and the Epithelial Barrier

The gut microbiota and host create a mutually beneficial relationship that maintains gut epithelial barrier integrity and mucosal immunity homeostasis [5]. Gut homeostasis refers to the body’s maintenance of a natural gut environment. Any cellular component disruption can predispose hosts to infections or initiate an abnormal immune response that leads to disease progression [46,47].

The gut microbiota plays a significant role in modulating host neural and immune development, morphogenesis and disease resistance [5]. The gut commensal microbiota is a crucial component of the gut immune system, and it includes all microorganisms found in the GI tract like archaea, bacteria, eukaryotes, fungi and viruses [5,47]. The commensal microbiota is tightly regulated to ensure that it is beneficial and remains constant to the host. Bidirectional interactions between the commensal bacteria and gut epithelium are essential for gut homeostasis to be maintained [46].

### 4.1. Impairment of Gut Epithelial Integrity

Damage to the epithelial integrity of the gut has serious consequences, including bacterial translocation [19], which triggers proinflammatory immune responses [48]. OID induces bacterial translocation and inflammation, mainly due to a shift in butyrate-producing bacteria, such as *Faecalibacterium prausnitzii*. The integrity of the tight junctions between intestinal epithelial cells is maintained by proteins, preventing the barrier from leaking and separating the mucosa and lumen. This includes tight junction protein rearrangement, which increases permeability and bacterial translocation from the gut lumen [46]

### 4.2. Intestinal Function Modulation

The major functions of the gastrointestinal tract include digestion, motility, secretion, absorption, tolerance and immune surveillance. Morphine prevents bicarbonate and mucus secretion from bronchi and the intestinal epithelium. Opioid administration is associated with a variety of gastrointestinal symptoms, such as nausea, constipation, leaky intestinal barrier function, bloating and vomiting [47]. It reduces intestinal motility by impairing coordinated myenteric activity, resulting in a delayed transit time and potentially raising the risk of bacterial translocation. According to experimental data carried out in both human and mice studies, the administration of opioids results in increased intestinal barrier permeability, bacterial translocation, an increased risk of enteric infection, gut-derived sepsis and a dysregulated immune response [5]. 

### 4.3. Host Immune System Modulation

Opioids alter the microbiome, increasing the risk of disease exacerbation and enhancing the development of tolerance to opioids [46]. In non-human primates such as apes, morphine produces immunosuppressive effects, slows T-cell maturation, alters cytokine secretion, and reduces the production of protein mediators of energy metabolism, signalling, and cell structure maintenance [5,19]. Morphine-induced neuro-immune interactions have immediate functional implications in the intestine. Morphine promotes an alteration of the function of the intestinal epithelium [27], and damages tight junction protein organisation via the toll-like receptor (TLR)-dependent modulation of myosin light-chain kinase [5].

Due to the constant barrage of pathogenic stimuli that impact the intestinal wall, the GI mucosa contains many immune cells; and there is substantial evidence that suggests that chronic opioid use reduces innate immunity [5,46]. Disruption in any cellular component can predispose the hosts to bacterial infection, microbial sensitization or the initiation of an abnormal immune response that accelerates disease progression [46,47].

### 4.4. Epithelial Barrier Function

There is also mounting evidence that neuroimmune interactions help maintain epithelial barrier integrity. Chronic opioid exposure disrupts gut homeostasis via various neuro-immune-epithelial mechanisms, leading to analgesic tolerance, which majorly underpins the opioid crisis. Differences in the molecular mechanisms of opioid tolerance between the enteric and central pain pathways pose a significant challenge for managing chronic pain without gastrointestinal side effects [46]. 

OID mechanisms may be linked to the disruption of epithelial barrier function as opioids suppress the immune system directly and promote inflammatory responses via interactions with enteric glia. An alteration in bacterial composition promotes barrier disruption, resulting in a “leaky” gut, and as a result, disease severity rises, and tolerance develops [47]. 

### 4.5. Morphine and the Gut Microbiome and Metabolome

Disruption of barrier function enables intestinal microbiota to translocate through the epithelium, which increases the susceptibility to infection by gut pathogens and faster infectious disease progression. In one study, two groups of mice were implanted with a 75 mg morphine pellet or a placebo pellet. After 24 h, mesenteric lymph node (MLN) (*n* = 9) and liver (*n* = 10) suspensions were collected and cultured overnight on blood agar plates (BD Biosciences), and the colony-forming units (CFUs) were measured. Mice receiving morphine revealed a higher number of CFUs than placebo-implanted mice, suggesting bacterial dissemination to MLN and liver after 24 h of morphine treatment [49]. 

The study further confirmed that the disseminated bacteria were from the gut lumen rather than from opportunistic infections. Mice were gavaged with ampicillin-resistant *E. coli* and quantified bacterial translocation with Lysogeny broth (LB) plates containing ampicillin. The results indicated that morphine treatment promotes bacterial translocation of commensal bacteria from the gut lumen as the mice showed ampicillin-resistant *E.coli* dissemination into the MLN and liver. Furthermore, morphine treatment enhanced the translocation of dextran coupled with fluorescein isothiocyanate (FITC) from the gut lumen to the blood, demonstrating that morphine increased the permeability of the gut epithelium [49]. 

In order to identify the functional changes in the gut microbiome, it is essential to identify the gut metabolomic profile. Morphine metabolism and elimination are vital in determining drug pharmacokinetics and evaluating drug efficacy and side effects. One study explored how morphine treatment changes the gut metabolomic profile by collecting faecal samples from the same animal on day zero, day one, day two and day four post-treatment, which were analyzed using chromatography-mass spectrometry [50].

The findings show that morphine treatment led metabolites to gradually and differentially alter over time. It also revealed a distinct clustering in the metabolomics profile compared to that observed after placebo treatment. As a result of morphine treatment, there is a decrease in bile, whereas phosphatidylethanolamines (PEs) and saturated fatty acids increased [50]. 

## 5. Inflammation and Infection

Drug dependence, including opioids, is associated with inflammation. Opioids can promote the release of pro-inflammatory cytokines from immune cells, which are involved in the upregulation of inflammation [22]. The central amygdala is the area of the brain where these inflammatory responses are recognised due to its behavioural and emotional drug-related stimuli role.

The use of opioids for their analgesic actions can have varying effects depending on the length of time they are used. Long-term opioid use can be correlated with decreased effectiveness in pain management [51]. In conditions associated with chronic pain, prescription opioids can eventually cause chronic inflammation. Research shows that the cause of this is the body’s response to inflammation: antibody creation [38]. 

### 5.1. Chronic Opioid Use and Immunosuppression

A study of patients suffering from chronic lower back pain showed that antibodies against opioids were found in chronic opioid users. In contrast, those using over-the-counter medications as an alternative developed no immunity towards these. The study also suggests that the antibody response correlates with the dosage of opioids [38]. 

Opioid use can also be linked to various infections through the immunosuppressive roles induced if used long-term. Research suggests that regular opioid users, who abuse the drug, have a higher prevalence of infections that solely affect immunocompromised patients. Research also reveals that patients that take opioids compliantly have been shown to have latent viruses reactivate or develop during obstetric medical treatment due to reduced immune function [52]. 

### 5.2. Human Immunodeficiency Virus (HIV)

It is common for opioid abusers to become infected by Human Immunodeficiency Virus (HIV), which damages immune cells, causes immunosuppression, and potentially progresses to acquired immune deficiency syndrome (AIDS), an umbrella term for several life-threatening infections and illnesses due to a damaged immune system by HIV. Opioids have been associated with aiding the development of HIV in the CNS and worsening the neurodegenerative diseases that are caused by chronic HIV [53]. 

### 5.3. Hepatitis C Virus (HCV)

Opioid abusers may be regularly involved in needle use, sharing and hazardous disposal, and unsafe sex (Table 2). These actions can easily expose them to HIV and the blood-borne hepatitis C virus (HCV). Research shows that opioids can speed up the development of HVC and cause chronic HVC by activating the opioid receptors found on immune cells [52]. The immune response to the influenza virus has also been shown to be compromised due to opioid usage. The influenza virus is a common cause of respiratory tract infections, scaling from mild upper respiratory infections to severe pneumonia. Opioids weaken the immune and inflammatory response leading to prolonged clearance of influenza through immunosuppressive actions previously discussed [52]. 

### 5.4. Herpes Simplex Virus

Herpes simplex virus (HSV) covers a range of infectious agents which cause oral and genital lesions, encephalitis, infections in neonates and malignant growths. Due to their immunosuppressive effects, opioids delay the HSV clearance, alter the virus itself and reactivate latent HSV [52]. Together, this demonstrates that immunosuppression by opioids reduces the body’s ability to fight infections and can exacerbate chronic conditions caused by infections.

## 6. Gastrointestinal Motility and Pathophysiology of the Gut by Opioids

As discussed above, opioids influence the GI tract by decreasing motility and peristalsis by slowing gastric emptying and increasing resting smooth muscle tone [27]. These factors contribute to OIBD.

Increased GI transit time has been shown to cause significant changes in the gut microbiome in constipation-predominant IBS. There are suggestions that similar changes with long-term opioid use could be driving the development of prolonged antinociception tolerance, meaning that they block the detection of painful stimuli [54].

The μ (MOR) and δ-opioid receptors (DOR) are the predominant opioid receptors in the GI tract [55]. MORs in enteric neurons are involved in the decrease in gut motility. When these receptors are activated, they cause a reduction in the excitability of the neurons and, subsequently, this slows gut motility [56].

The MORs are primarily found in myenteric and submucosal plexuses. Their importance in the modulation of gut motility is evident, as this effect is not shown in μ-opioid receptor knockout mice [57]. Activation of the MORs on myenteric neurons affects gut motility, as these neurons are responsible for the innervation of smooth muscle. Acetylcholine (ACh) stimulates longitudinal smooth muscles in the gut, and vasoactive intestinal peptide (VIP) and nitric oxide mediate the inhibition of inhibitory neurons in the circular muscles. Opioids prevent neurotransmitter release and therefore bring about the effects of increased tone and reduction of normal peristaltic activity [58]. 

Stimulation of the DORs and MORs on submucosal neurons causes a decrease in secretion and absorption due to the release of ACh and VIP being inhibited. Cl^−^ secretion is promoted while the Na^+^/Cl^−^ absorption is inhibited. This disturbs the water-electrolyte balance, which is vital for digestion and resistance to bacterial infections [43].

Opioids decrease enteric neuronal excitation and consequently inhibit neurotransmitter release by altering ion channels. An increase in potassium results in the membrane being hyperpolarized and therefore halts the action potential generation. Sodium and calcium channels are suppressed. Morphine causes sodium channels to be inhibited, thus reducing excitation in the neurons because they cannot reach the threshold for multiple action potential firing [59]. 

### 6.1. Opioid-Induced Bowel Dysfunction (OIBD)

OIBD refers to the gastrointestinal symptoms associated with the use of opioids. These appear due to slowing intestinal motility, uncoordinated contractions and increased sphincter tone. Symptoms include ‘gastroesophageal reflux, vomiting, bloating, abdominal pain, anorexia, and constipation’ [55]. Of these, opioid-induced constipation seems to be of greatest clinical importance and links to upper GI symptoms.

### 6.2. Opioid-Induced Constipation (OIC)

OIC is described as altered bowel habits and defecation after initiating opioid therapy and is characterised by a reduced bowel movement, exacerbated straining to pass a bowel movement, feeling of incomplete evacuation and/or harder stools [60]. 

Opioids reduce neurotransmitter release by binding to submucosal secretomotor neurons, causing a reduction in chlorine secretion. Consequently, chlorine no longer maintains the osmotic gradient required to allow water into the intestinal lumen and therefore, contributes to OIC. The time in which water can be absorbed is prolonged due to the reduction of gut motility. As water is absorbed, faecal volume is reduced. Peristaltic activity relies on mechanoreceptor activation, and so the decrease in volume further impacts gut motility and OIC [61].

Another factor that might promote constipation with opioids is that opioids can increase sphincter tone and cause muscle spasms. Those on opioid therapy become tolerant to the analgesic effects of opioids but not to the GI effects [62]. In colonic neurons, β-arrestin-2 prevents tolerance to GI effects, whereas it has the opposite effect in central neurons, where it mediates tolerance.

When opioids bind to MORs G-signalling leads to phosphorylation of the receptor, bringing about β-arrestin-2 recruitment [63]. The significance of this molecule has been noted, as β-arrestin-2 knockout mice showed tolerance in the colon [18]. 

With chronic opioid use, there is downregulation of β-arrestin-2 in the ileum, whereas in the colon, β-arrestin-2 continues to be expressed. This explains the difference in tolerance between the ileum and colon, where the ileum becomes tolerant to opioid effects, and in contrast, the colon does not. Receptor recycling occurs in the colon, manifesting as the continued activation of receptors and progressing to OIC [56]. 

## 7. Therapeutic Options and Treatment Perspectives

Opioid-related GI and neurological disorders are presented in various ways, and clinicians must be aware of the different presentations and treatments available for these side effects.

### 7.1. Faecal Microbiota Transplantation (FMT)

Faecal Microbiota Transplantation (FMT) is the administration of the entire microbial community from the stool of a healthy donor into the intestinal tract of the recipient in order to normalise or modify the composition and function of the intestinal microbiota [64]. In a recent study to combat the effect of OID, the findings suggest a potential role for the gut microbiome in expressing the somatic signs of morphine withdrawal, which might improve opioid dependence and withdrawal therapies as defined by quantification of naloxone-precipitated withdrawal jumps [65]. The findings suggest that FMT restores and improves morphine-treated recipient mice microbial communities. In addition, morphine-treated animals receiving FMT from morphine-treated donor mice showed lower levels of naloxone-precipitated opioid withdrawal [65]. 

### 7.2. Antibiotic Treatment

High antibiotic doses have been shown to result in a more significant levels of microbial clearance. The antibiotics treatment regimen produced a robust suppression of naloxone-precipitated opioid withdrawal in morphine-dependent mice [65]. However, in another study, no changes in naloxone-precipitated jumping were observed in morphine-pelleted mice after antibiotic treatment, owing to differences in morphine dosing regimens between studies [66]. 

The antibiotics used in this investigation are neomycin, vancomycin and metronidazole, with a 10-fold increase in dosage compared to the above studies. Antibiotics have been demonstrated to affect the structure and function of the nervous system directly. However, the mechanism is still unclear [65]. 

### 7.3. Probiotic & Prebiotic Therapy

Probiotics are live, nonpathogenic microorganisms that improve microbial balance, especially in the GI tract [67]. During morphine administration, probiotic therapy should be considered. Counteracting opioid-induced increased gut permeability and neuroinflammation may provide a way to prolong morphine’s efficacy while reducing side effects [65]. Some examples are listed in Table 1 below.

**Table 1 biomedicines-10-01815-t001:** Probiotics and their function.

Probiotics	Function
*Bifidobacteria* and *Lactobacillaeae*	Tolerant to morphine, and probiotics containing these bacterial communities can prevent the development of analgesic tolerance in the morphine-treated rats [66].
*B. longum* and *L. rhamnosus*	Maintain a healthy intestinal barrier [68] bacterial translocation, and neuroinflammation by manipulating the gut microbiome [69].
*Lactobacillus genus*	In mice, it was shown to induce opioid and cannabinoid CB2 receptor expression and mediate analgesic activities in intestinal epithelial cells [70].

Prebiotics are non-viable food ingredients that are selectively metabolised by intestinal bacteria. Prebiotic dietary modulation of the gut microflora is intended to improve health by increasing the number and/or activity of bifidobacteria and lactobacilli [40]. Fructooligosaccharides are considered to be prebiotics and influence ENS function by modulating the gut microbiota. Chronic treatment with fructooligosaccharide prebiotics in diabetic mice fed a high-fat diet (45%) leads to a decrease in body weight associated with a decrease in fasting hyperglycemia [71]. 

### 7.4. Myosin Light Chain Kinase (MLCK) Inhibitor ML-7

The impact of the inhibitor ML-7, shown in earlier research, is also worth noting as it protects the barrier function of several endothelial and epithelial cell lines [49]. Previous studies observed that the gut epithelial barrier was protected against disruption in morphine-pelleted mice and inhibited morphine-induced bacterial translocation [72]. ML-7 might be involved in mechanisms other than gut permeability regulation, hence additional research is required.

### 7.5. Opioid-Induced Constipation (OIC)

The most common and widely reported adverse effect of opioid use is OIC, which affects 40–80% of opioid users. After a diagnosis of OIC is made, physicians can use the bowel function index (BFI) to evaluate patients’ symptoms and ask patients to rank the following symptoms from 0 (not at all) to 100 (extremely severe) in the previous seven days: a feeling of incomplete bowel evacuation, ease of defecation, and patient’s judgement of constipation [72]. 

Although laxatives are recommended as first-line agents for OIC treatment, they do not relieve OIC symptoms in all patients [73]. They can cause side effects such as flatulence, bloating, and a sudden urge to defecate, interfering with daily activities. As a result, several new and more specific pharmacological approaches are emerging [74]. 

Several treatment guidelines recommend that Peripherally Acting μ Opioid Receptor Antagonists (PAMORAs) should be considered when starting opioid therapy or in patients with OIC who do not respond to laxatives. One of the most suitable PAMORAs for daily administration is oral Naloxegol. In two randomised, placebo-controlled phase 3 trials, naloxegol was considerably more effective than placebo in the overall patient population. In both studies, patients classified as laxative-inadequate responders were given 25 mg, and in one study, 8.5 mg [75]. 

Relying on one treatment for opioid-induced dysbiosis in acute and chronic pain is insufficient. Opioids have immunosuppressive effects in peripheral immune cells but pro-inflammatory effects in the CNS. As a result, vaccines have emerged as one of the most promising preventative/therapeutic alternatives to combat Opioid Use Disorder (OUD). The vaccine will produce opioid-specific antibodies that bind to the target opioid, limiting drug-induced behavioural and pharmacological effects and lowering drug distribution to the brain [75]. Anti-opioid vaccines can be used in conjunction with FDA-approved medications due to their selectivity. However, the current research has several limitations, such as being limited to male mice and rats and the Specific Pathogen-Free (SPF) condition did not accurately capture the microbiome changes seen in chronic opioid users [76]. Hence, more research is needed to examine the relationship between the microbiome and vaccination against OUD.

Another approach to combat OUD is Quantitative Systems Pharmacology (QSP). A QSP technique studies the dynamic interactions between drugs and a biological system quantitatively, allowing for a better understanding of the system’s behaviour rather than the individual components. Understanding how different components of the biological system interact can help researchers find biomarkers to predict disease severity and treatment outcomes [77]. 

Chronic opioid treatment causes a substantial loss in gut barrier integrity and bacterial migration. The loss of gut barrier integrity can exacerbate viral infection and sepsis, which has serious clinical implications given the propensity to use opioids in emergency settings. Following chronic opioid therapy, it has been demonstrated that both Gram-positive and Gram-negative bacteria invade the liver, spleen and lymph nodes. Restoring the gut flora is a clear target for improving opioid efficacy and reducing withdrawal symptoms [78]. 

**Table 2 biomedicines-10-01815-t002:** Summary of microbials related to the gut.

Microbial	Effect on the Gut
*Mycobacterium*	Major pathogen of IBD [26]
*Escherichia coli*	Major pathogen of IBD [26]
*Clostridium difficile*	Major pathogens of IBD [26]
Lactobacillaceae	Association with Parkinson disease [27]
Enterococcaceae	Association with Parkinson disease [27]
Alistipes	Increases prevalence in depression [78]
*Faecalibacterium prausnitzii*	Has anti-inflammatory properties that promote gut health. Decrease in this bacteria can have consequences on the epithelial integrity of the gut [46].
Human Immunodeficiency Virus (HIV)	HIV is a virus that attacks its own immune system. Opioids have been associated with aiding the development of HIV in the CNS and worsening the neurodegenerative diseases that are caused by chronic HIV [53].
Hepatitis C virus (HCV)	Opioid abusers may be regularly involved in needle use, sharing and hazardous disposal, and unsafe sex. These actions can easily expose them to HIV and blood-borne HCV [52].
Herpes Simplex Virus	Herpes simplex virus (HSV) covers a range of infectious agents which cause oral and genital lesions, encephalitis, infections in neonates and malignant growths. Opioids delay the HSV clearance, alter the virus itself and reactivate latent HSV [52].
Bifidobacterium genus	Probiotics that normally live in the gut [22].

## 8. Future Directions and Conclusions

Current studies indicate the impact of change in the gut microbiota with opioids and their role in inflammation, immunomodulation and damage to the epithelial barrier [79]. Many have documented a link between dysbiosis and analgesic tolerance through pro-inflammatory signalling pathways [79]. Recent findings imply the significance of pain pathways in contributing to tolerance, with evidence suggesting the involvement of primary afferent neurons [80,81]. Further research into the specific role of the microorganisms associated with OID will improve understanding of the correlation between these processes and antinociceptive tolerance and aid the development of novel treatments.

OIH is another area of interest, with studies demonstrating that hyperalgesia reduces the potency of opioids. As a result, an increase in the dose of opioids brings about issues such as addiction, dependence and withdrawal [82]. Whether there is a relationship between OID and OIH is yet to be evaluated.

The involvement of the gut-brain axis seems to be of increasing importance in inducing antinociceptive tolerance. Dysbiosis and associated immune dysregulation point towards the progression of neurological problems and OUDs [83,84], whereby reduced GI motility and OIC appear to worsen these effects.

More research is necessary to determine the correlation between microbial dysbiosis and the side effects of opioids. Further research should explore how opioids actively change the microbial make-up of the gut, but also how this results in the side effects seen in opioid use.

The symptoms associated with opioid use, such as OIBD and mental health issues- in addition to OIH and OUDs, highlight the need for new treatments that minimise the dose and duration of opioid therapy. Future research into FMT, a potential anti-opioid vaccine, prebiotics and probiotics is required to prove their effectiveness in alleviating and preventing opioid-induced side effects, meanwhile reducing the risk of tolerance. More research involving humans needs to be done, as much of the work is focused only on animals.

There are now numerous studies on OID, but very few consider the difference in the effects and management of OID in acute compared to chronic pain. Additional research is necessary to differentiate how acute versus chronic opioid use influences the microbiome and the gut-brain axis and if this should impact recommended treatments.

## Data Availability

Not applicable.

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
