# Peer review of "Pain and Opioid-Induced Gut Microbial Dysbiosis"

_biomedicines, 2022, doi:10.3390/biomedicines10081815_

Round 1

Reviewer 1 Report

It would be an interesting manuscript if some improvements will be addressed:

1. I wanted to have a graphical abstract to gain more readers

2. Potential antinociceptive tolerance to opioids through changes in the gut microbiota that need to be mentioned

3. Some pathophysiology of the gut by opioids needs to be further involved.

4. An additional chapter about how morphine induces changes in the gut microbiome and metabolome should be provided 

5. In order to gain more understanding of the overall manuscript, a table aims to summarize all microbial related to the Pain and Opioid-Induced that need to be provided

6. I wanted to add a chapter about the involvement of the gut-brain axis and treatment regards chronic opioids should be discussed

Author Response

  1. We are more than happy to provide a graphical abstract but will need more time. We propose to send it later not to delay the submission of this revised version of our manuscript.

2. The following paragraphs have been added to further highlight this: 

Given the clinical and social implications of physical dependence and addiction, the mechanisms causing opioid tolerance have been studied for many decades. Despite multiple studies at various levels in different tissues, no single regulatory mechanism can account for all of the variations observed in tolerance development. This shows how complex a phenomenon opioid tolerance is, as it involves modulation by multiple cellular pathways.

Antinociceptive tolerance is clinically manifested as decreased or lost pain relief from a given opioid dose administered repeatedly or with continuous administration of an opioid over a period of time. The need for increasing opioid doses in chronic pain cases is well documented and is typically presented as a major barrier to providing adequate pain relief over a long period of time. 

Chronic morphine has recently been shown to alter the gastrointestinal tract's resident microbial composition and induce bacterial translocation across the gut epithelial barrier via mechanisms involving toll-like receptors (TLR) 2 and 4.

The primary mechanism for tolerance to the antinociceptive effects of chronic morphine appears to be dysbiosis of the gut microbiome. The precise mechanism by which the gut microbiome influences tolerance development is unknown; however, it is possible that breakdown of the epithelial barrier, microbial translocation, and inflammation within the colonic wall may alter extrinsic sensory neurons, resulting in tolerance development.

3. Section 6 now involves gastrointestinal motility and pathophysiology. 

4. Disruption of barrier function enables intestinal microbiota to translocate through the epithelium, which increases the susceptibility to infection by gut pathogens and faster infectious disease progression. In one study, two groups of mice were implanted with 75 mg morphine pellet or a placebo pellet. After 24 hours, mesenteric lymph node (MLN) (n = 9) and liver (n = 10) suspensions were collected and cultured overnight on blood agar plates (BD Biosciences), and the colony-forming units (CFUs) were measured. Mice receiving morphine revealed a higher number of CFUs than placebo-implanted mice, suggesting bacterial dissemination to MLN and liver after 24 hours of morphine treatment. 

The study further confirmed that the disseminated bacteria were from the gut lumen rather than opportunistic infections. Mice were gavaged with ampicillin-resistant E.coli and quantified bacterial translocation with Lysogeny broth (LB) plates containing ampicillin. The results indicated that morphine treatment promotes bacterial translocation of commensal bacteria from the gut lumen as the mice showed ampicillin-resistant E.coli dissemination into MLN and liver. Furthermore, morphine treatment enhanced the translocation of dextran coupled with fluorescein isothiocyanate (FITC) from the gut lumen to the blood, demonstrating that morphine increased the permeability of the gut epithelium. 

In order to identify the functional changes in the gut microbiome, it is essential to identify the gut metabolomic profile. Morphine metabolism and elimination are vital in determining drug pharmacokinetics and evaluating drug efficacy and side effects. One study explored how morphine treatment changes the gut metabolomic profile by collecting faecal samples from the same animal on day 0, day 1, day 2 and day 4 post-treatment, which were analyzed using chromatography-mass spectrometry.

The findings show that morphine treatment led metabolites to gradually and differentially alter over time. It also revealed a distinct clustering in the metabolomics profile compared to that observed after placebo treatment. As a result of morphine treatment, there is a decrease in bile, whereas phosphatidylethanolamines (PEs) and saturated fatty acids increased.

5. â€‹The following table has been added: 

Microbial

Effect on the Gut 

Mycobacterium 

Major pathogen of IBD

Escherichia coli 

Major pathogen of IBD

Clostridium difficile 

Major pathogens of IBD

Lactobacillaceae 

Association with Parkinson disease 

Enterococcaceae 

Association with Parkinson disease 

Alistipes

Increases prevalence in depression 

Faecalibacterium prausnitzii

Has anti-inflammatory properties that promote gut health. Decrease in this bacteria can have consequences on the epithelial integrity of the gut. 

Human Immunodeficiency Virus (HIV)

HIV is a virus that attacks its own immune system. Opioids have been associated with aiding the development of HIV in the CNS and worsening the neurodegenerative diseases that are caused by chronic HIV.

Hepatitis C virus (HCV)

HCV is a liver infection. Opioid abusers may be regularly involved in needle use, sharing and hazardous disposal, and unsafe sex. These actions can easily expose them to HIV and blood-borne HCV. 

Herpes Simplex Virus 

Herpes simplex virus (HSV) covers a range of infectious agents which cause oral and genital lesions, encephalitis, infections in neonates and malignant growths. Opioids delay the HSV clearance, alter the virus itself and reactivate latent HSV

Bifidobacterium genus

Are probiotics that normally live in your gut. 

6. Section 3 shows the involvement of the gut-brain axis and treatment regarding chronic opioids has been added as below: 

Chronic opioid treatment causes a substantial loss in gut barrier integrity and bacterial migration. The loss of gut barrier integrity can exacerbate viral infection and sepsis, which has serious clinical implications given the propensity to use opioids in emergency settings. Following chronic opioid therapy, it has been demonstrated that both Gram-positive and Gram-negative bacteria invade the liver, spleen, and lymph nodes. Restoring the gut flora is a clear target for improving opioid efficacy and reducing withdrawal symptoms. 

Reviewer 2 Report

In the manuscript titled "Pain and Opioid-Induced Gut Microbial Dysbiosis" Karen Raechel Thomas and colleagues, they have reported that opioid-induced dysbiosis (OID) is a specific condition describing the consequences of opioid use on the bacterial composition of the gut. Opioids have been shown to affect the epithelial barrier in the gut and modulate inflammatory pathways, possibly mediating opioid tolerance or opioid-induced hyperalgesia; in combination, these allow the invasion and proliferation of non-native bacterial colonies. There is also evidence that the gut-brain axis is linked to the emotional and cognitive aspects of the brain with intestinal function, which can be a factor that affects mental health. For example, Mycobacterium, Escherichia coli, and Clostridium difficile are linked to Irritable Bowel Disease; Lactobacillaceae and Enterococcacae have associations with Parkinson’s disease, and Alistipes has increased prevalence in depression. However, changes to the gut microbiome can be therapeutically influenced with treatments such as fecal microbiota transplantation, targeted antibiotic therapy, and probiotics. There is also evidence of emerging therapies to combat OID. This review has collated evidence that shows there are correlations between OID and depression, Parkinson’s disease, infection, and more. Specifically, in pain management, targeting OID deserves specific investigations. I have a few comments regarding the present manuscript.

-Title, please add more information in the affiliations (2, 3, 4)

-Introduction, please check the format of the references in the entire document.

-"The impact of opioids on the balance and composition of gut microbiota has recently received increased attention". what papers support this sentence?

-Reading the introduction, maybe a better link between opioids and gut microbiota is required.

-"According to recent research, cholecystokinin is upregulated in the rostral ventromedial medulla (RVM) during chronic opioid exposure. Cholecystokinin has anti-opioid and pronociceptive properties, activating descending pain facilitation mechanisms from the RVM, enhancing nociceptive transmission at the spinal cord, and promoting hyperalgesia.

The neuroplastic adaptive changes induced by opioid exposure promotes increased pain transmission, resulting in decreased antinociception (i.e. tolerance)12", the authors have stated that recent research was made, and the reference is from 2004. A long paragraph maybe requires more references.

-Experimental evidence section, .... "several experiments have been carried out" and the authors have referenced only two papers from 2006 and 2009. 

-Maybe some abbreviations require a previous definition e.g. GI and CNS

-What is the importance of section 4 in this review?

-Reading the sections 3, 4, and 5, all the paragraphs are supporting for singular or only two references, why this method?

-Why gastrointestinal motility and constipation is a chapter of the present review?. because of the effects of opioids?

-Section 7 is very similar to section 3, maybe mixing these two sections is a good idea.

-Sections 8 and 9 were the best in my opinion. Bifidobacteria is not italics. 

The manuscript requires a profound revision in the main idea, why this OID is so important, and explain the impact on the health, later the gut-brain axis, and the blood-brain barrier. Treatments, and further directions. Finally, supporting all the paragraphs with more than a singular paper, using always original papers, not review papers.

Author Response

1. More information has been added to the affiliations as follows: 

1- University of Aberdeen, School of Medicine, Medical Sciences and Nutrition, Foresterhill, Aberdeen, Scotland, AB25 2ZD, UK 

2- Rowett Institute, University of Aberdeen, Foresterhill, Aberdeen, Scotland, AB25 2ZD, UK

3- Institute of Applied Health Sciences, Epidemiology Group,  University of Aberdeen, Foresterhill, Aberdeen, Scotland, AB25 2ZD, UK

4- NHS Grampian, Department of Anaesthesia, Foresterhill, Aberdeen, Scotland, AB25 2ZD, UK

2. The references have been edited and adjusted according to the journal’s requirements. 

3. This sentence has been removed from the paragraph. 

4. The following sentence in the abstract aims to highlight the link between opioids and gut microbiota. 

Opioid-Induced Dysbiosis (OID) is a specific condition describing the consequences of opioid use on the bacterial composition of the gut. Opioids alter the microbiome by reducing gastrointestinal transit through the effects on μ-opioid receptors. Opioids have been shown to affect the epithelial barrier in the gut and modulate inflammatory pathways, possibly mediating opioid tolerance or opioid-induced hyperalgesia; in combination, these allow the invasion and proliferation of non-native bacterial colonies.

5. More references have been added to support this statement. 

  1. Marshall, T.M.; Herman, D.S.; Largent-Milnes, T.M; Badghisi, H.; Zuber, K.; Holt, S.C.; Lai, J.; Porreca, F.; Vanderah, T.W. (2012)  Activation of descending pain- facilitatory pathways from the rostral ventromedial medulla by cholecystokinin elicits release of prostaglandin- e2 in the spinal cord. Pain, 153(1), 86-94.
  2. Meng, I. D.; Harasawa, I. (2007). Chronic morphine exposure increases the proportion of on-cells in the rostral ventromedial medulla in rats. Life sciences, 80(20), 1915-1920.

6. References have been added to further support this statement.  

1. Angst, M. S.; Clark, J. D. (2006). Opioid-induced hyperalgesia: a qualitative systematic review. The Journal of the American Society of Anesthesiologists, 104(3), 570-587.

2. Arner, S.; Rawal, N.; Gustafsson, L. L. (1988). Clinical experience of long–term treatment with epidural and intrathecal opioids–a nationwide survey. Acta anaesthesiologica Scandinavica, 32(3), 253-259.

3. Sjøgren, P.; Thunedborg, L. P.; Christrup, L.; Hansen, S. H.; Franks, J. (2008). Is development of hyperalgesia, allodynia and myoclonus related to morphine metabolism during long-term administration?: Six case histories. Acta anaesthesiologica scandinavica, 42(9), 1070-1075.

7. The following text has been added to explain the terms GI and CNS in the introduction. 

The gastrointestinal (GI)  system consists of the organs that form the gastrointestinal tract from the mouth to the anus. 

The central nervous system (CNS) consists of the brain (encephalon) and the spinal cord (medulla spinalis) and is the body’s processing centre. 

8. Section 4 aims to show the link between opioid-induced dysbiosis and epithelial barrier function and how opioids can affect this barrier. 

9. A few more original references have been added to ensure that there are a minimum of more than 4 references per section. 

10. Yes, this chapter was added to highlight the effects opioids have on the gut and how they can cause constipation and gut dysfunction. 

11. We have put section 7 under section 3 

12. Bifidobacteria has been changed from italics to the normal font. 

13. More references have been added to the relevant sections to further support it.

Reviewer 3 Report

In this interesting review, authors collected information regarding gut microbial dysbiosos that is induced with opioids. Review is well conducted, however, I have a few comments that should be addressed:

- adjust references to be in line with journal's guidelines

- subchapters 1 and 2 could be shorter, as they seem not to be critically connected to the main topic of the review

- in my opinion, figure 1 is not necessary, especially as it is only adapted from different manuscript. Instead, authors should incorporate original figure, or additional table that describe key conclusions from this review

Author Response

1. The references have been adjusted and edited according to the journal’s guidelines. 

2. Some parts of subchapters 1 and 2 have been edited to make them shorter.

3. Fig 1 has been deleted and a table that summarises all the microbials related to Pain and Opioid-Induced Dysbiosis has been added. 

Microbial

Effect on the Gut 

Mycobacterium 

Major pathogen of IBD

Escherichia coli 

Major pathogen of IBD

Clostridium difficile 

Major pathogens of IBD

Lactobacillaceae 

Association with Parkinson disease 

Enterococcaceae 

Association with Parkinson disease 

Alistipes

Increases prevalence in depression 

Faecalibacterium prausnitzii

Has anti-inflammatory properties that promote gut health. Decrease in this bacteria can have consequences on the epithelial integrity of the gut. 

Human Immunodeficiency Virus (HIV)

HIV is a virus that attacks its own immune system. Opioids have been associated with aiding the development of HIV in the CNS and worsening the neurodegenerative diseases that are caused by chronic HIV.

Hepatitis C virus (HCV)

HCV is a liver infection. Opioid abusers may be regularly involved in needle use, sharing and hazardous disposal, and unsafe sex. These actions can easily expose them to HIV and blood-borne HCV. 

Herpes Simplex Virus 

Herpes simplex virus (HSV) covers a range of infectious agents which cause oral and genital lesions, encephalitis, infections in neonates and malignant growths. Opioids delay the HSV clearance, alter the virus itself and reactivate latent HSV

Bifidobacterium genus

Are probiotics that normally live in your gut. 

Round 2

Reviewer 2 Report

Thanks for your best wishes Thanks again for giving me the track-changes paper Reading the manuscript I found that the authors have followed all my previous requests that were not visualized in the platform in the previous revision. No further comments are required from my side, my decision is accepted with minor changes.